# Benchmarking Large Vision-Language Models on Fine-Grained Image Tasks: A Comprehensive Evaluation

**Hong-Tao Yu**[1], **Yuxin Peng**[2], **Serge Belongie**[3], **Xiu-Shen Wei**[1*]

[1]School of Computer Science and Engineering, School of Intelligence Science and Engineering and Key Laboratory of New Generation Artificial Intelligence Technology and Its Interdisciplinary Applications, Southeast University, China
[2]Wangxuan Institute of Computer Technology, Peking University, China
[3]University of Copenhagen, Denmark

## Abstract

Recent advancements in Large Vision-Language Models (LVLMs) have demonstrated remarkable multimodal perception capabilities, garnering significant attention. While numerous evaluation studies have emerged, assessing LVLMs both holistically and on specialized tasks, fine-grained image tasks—fundamental to computer vision—remain largely unexplored. To fill this gap, we introduce a comprehensive fine-grained evaluation benchmark, *i.e.*, `FG-BMK`, comprising 1.01 million questions and 0.28 million images. Our evaluation systematically examines LVLMs from both human-oriented and machine-oriented perspectives, focusing on their semantic recognition and fine-grained feature representation capabilities. Through extensive experiments on twelve representative LVLMs/VLMs, we uncover key findings regarding the influence of training paradigms, modality alignment, perturbation susceptibility, and fine-grained category reasoning on task performance. This work provides critical insights into the limitations of current LVLMs and offers guidance for future data construction and model design in the development of more advanced LVLMs. Our code is open-source and available at `https://github.com/SEU-VIPGroup/FG-BMK`.

## 1 Introduction

Large language models have recently achieved remarkable advancements, with models like GPT-4 (OpenAI, 2023) surpassing human-level performance across diverse tasks. Building on this, Large Vision-Language Models (LVLMs) have rapidly evolved, with models such as GPT-4o (OpenAI, 2023), Qwen (Bai et al., 2023), InternVL (Chen et al., 2024), and LLaVA-1.5 (Liu et al., 2024a) demonstrating strong multimodal reasoning and perception capabilities.

In response to these advancements, various holistic and specialized evaluations have emerged to assess the capabilities of LVLMs. For instance, LVLM-eHub (Xu et al., 2025) and MMBench (Yuan et al., 2024) provide broad evaluations of multimodal perception and reasoning, while specialized evaluations like DocVQA (Mathew et al., 2021) and GQA (Hudson & Manning, 2019) focus on specific tasks, such as document visual perception and visual reasoning. While these evaluations have provided valuable insights, a critical gap remains: there is no systematic evaluation of LVLMs on fine-grained image tasks, which focus on analyzing visual objects at the subordinate category level and are fundamental to computer vision (Wei et al., 2022b). Consequently, the capability boundaries of LVLMs in fine-grained tasks remain poorly understood.

---
*Corresponding author. This work was supported by National Natural Science Foundation of China under Grant (62522602, 62272231, 62525201, 62132001, 62432001), Basic Research Program of Jiangsu under Grant (BK20250073), Beijing Natural Science Foundation (L247006), the Pioneer Centre for AI, DNRF grant number P1, the Fundamental Research Funds for the Central Universities (4009002401, 2242025K30024), and the Big Data Computing Center of Southeast University.

To address this gap, we propose a comprehensive fine-grained evaluation benchmark, termed `FG-BMK`. This benchmark includes 1.01 million questions and 0.28 million images, providing a robust test bed for evaluating LVLMs. `FG-BMK` incorporates two evaluation paradigms: human-oriented and machine-oriented. The human-oriented evaluation employs dialogue-like interactions to assess a model's ability to understand and respond to fine-grained visual queries in a conversational context. The machine-oriented evaluation focuses on two core fine-grained vision tasks—image retrieval and recognition—to directly measure the feature representation capabilities of LVLMs. Together, these evaluations enable a comprehensive assessment of LVLMs' fine-grained feature representation and semantic recognition abilities.

Through extensive evaluations across representative LVLMs and VLMs within `FG-BMK`, we derive several key insights:

- The contrastive training paradigm in LVLMs proves more effective in enhancing the fine-grained discriminability of visual features, whereas generative and reconstruction-based training paradigms tend to yield weaker discriminability.
- Aligning visual features with textual features in LVLMs can impair their fine-grained discriminability, particularly when there is a mismatch in granularity between the paired image-text data.
- LVLMs and LVMs are more vulnerable to feature perturbations in fine-grained tasks than in generic vision tasks.
- LVLMs demonstrate relatively stronger capabilities in perceiving visual appearances but face challenges in fine-grained category reasoning (which depends on the recognition of visual attributes).
- Despite their advancements, LVLMs still lag behind fine-grained tailored models in handling fine-grained visual tasks.

These findings provide a foundation for future research aimed at advancing LVLM performance in fine-grained tasks and addressing the challenges uncovered by `FG-BMK`.

## 2 RELATED WORK

**Large Vision-Language Models** Large Language Models (LLMs) like GPT-4 (OpenAI, 2023) have made notable strides in text comprehension, reasoning, and generation. Building on this foundation, Large Vision-Language Models (LVLMs) have emerged with impressive reasoning and perception abilities across diverse tasks. Enhancing LVLM performance has taken various directions: BLIP (Li et al., 2022) uses noisy web data with bootstrapped captions; LLaVA (Liu et al., 2024a) employs GPT-4-generated instruction-following data; BLIP-2 (Li et al., 2023) integrates frozen image and text encoders; Qwen2.5-VL (Bai et al., 2023) leverages dynamic-resolution strategy for preserving native image resolution; InternVL3 (Zhu et al., 2025) unifies pre-training over multimodal data for improved effectiveness. While achieving success on many tasks, fine-grained visual challenges for these LVLMs remain underexplored. This work addresses this gap by assessing their performance in fine-grained domains, uncovering both their potential and limitations.

**Large Vision-Language Model Benchmarks** While LVLMs have shown impressive performance, various benchmarks have been developed to evaluate their capabilities in different domains. Specialized benchmarks like ChartQA (Masry et al., 2022) for chart understanding, GQA (Hudson & Manning, 2019) for visual reasoning, and CAPability (Liu et al., 2025) for caption evaluation provide focused assessments. Additionally, evaluations in optical character recognition (Liu et al., 2024c) and adversarial robustness (Madry et al., 2018) offer insights into specialized aspects of LVLM performance. Holistic evaluations, such as LVLM-eHub (Xu et al., 2025) and MMBench (Yuan et al., 2024), assess general multimodal perception and reasoning, while others like MathVista (Lu et al., 2024) and MMMU (Yue et al., 2024) include expert-level problems spanning multiple disciplines.

However, comprehensive evaluations of LVLMs in fine-grained visual tasks remain scarce. Existing efforts, such as (Geigle et al., 2024), primarily focus on fine-grained classification, and (Zhang et al., 2024c) evaluates classification and reasoning with limited questions. To address it, we propose `FG-BMK`, a benchmark designed to comprehensively evaluate LVLMs' fine-grained feature representation and semantic recognition capabilities across diverse visual tasks.

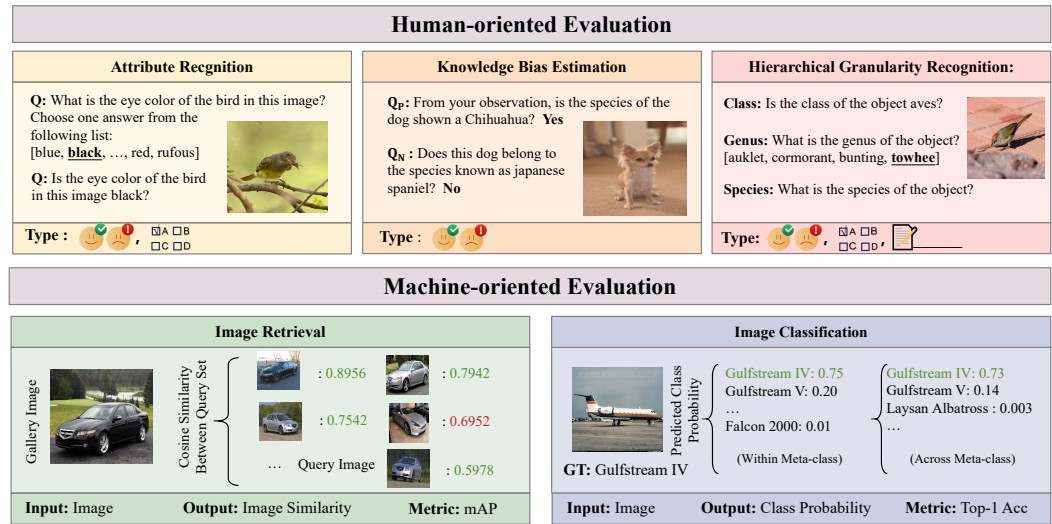

Figure 1: Our proposed benchmark: The human-oriented evaluation tests the model's ability to handle fine-grained visual queries (true/false, multiple-choice, short-answer), while the machine-oriented evaluation directly assesses visual feature representation through image retrieval and classification tasks. 🙂🙁=true/false question, ⊞=multiple-choice question, 📝=short-answer question.

**Fine-Grained Image Tasks** Fine-grained visual tasks (Wei et al., 2022b; Xu et al., 2022; Wei et al., 2024; Zhang et al., 2024a; Jing et al., 2024a; Wang et al., 2024; Liu et al., 2024b; Zhang et al., 2024b) are pivotal in applications like biodiversity monitoring (Jing et al., 2024b), object retrieval (Shen et al., 2022), and product recommendation (Wei et al., 2022a). While LVLMs such as GPT-4, InternVL, and Qwen excel in tasks like OCR, visual question answering, and image captioning, fine-grained tasks remain a significant challenge. These tasks demand models to identify subtle, discriminative patterns in images (Song et al., 2020) and leverage expert knowledge from LLMs for precise responses. To address this, we introduce a comprehensive benchmark and conduct extensive experiments to evaluate LVLMs' performance on fine-grained tasks. Our study systematically highlights their limitations and offers actionable insights for advancing model design and training.

## 3 THE EVALUATION BENCHMARK

### 3.1 EVALUATION TASKS AND METRICS

To comprehensively evaluate LVLMs, we adopt two evaluation paradigms: 1) *human-oriented evaluation* and 2) *machine-oriented evaluation*.

Specifically, since dialogue is the most direct way for humans to interact with LVLMs, the human-oriented evaluation assesses the model's ability to understand and respond to fine-grained visual queries in a conversational setting. We utilize three types of questions—true/false, multiple-choice, and short-answer (cf. Figure 1)—where a response is considered correct if it includes the ground truth.

While, the machine-oriented evaluation encompasses two fundamental vision tasks (Wei et al., 2022b): image retrieval and image classification. To evaluate the feature representation ability of LVLMs, we measure the accuracy of their visual features on these tasks. Following the DINOv2 approach (Oquab et al., 2023), we use mean Average Precision (mAP) for retrieval and Top-1 accuracy for classification.

The detailed aspects of both human-oriented and machine-oriented evaluations are outlined below:

***Human-oriented Evaluation***:

- *Attribute Recognition*: This includes true/false questions and multiple-choice questions, designed to evaluate the model's ability to identify visual attributes such as size, color, length, shape, pattern, and more.

- *Knowledge Bias Estimation*: This section consists exclusively of true/false questions, designed to uncover potential knowledge biases across different fine-grained categories by evaluating LVLM's accuracy for each category.
- *Hierarchical Granularity Recognition*: This section includes true/false questions, multiple-choice questions, and short-answer questions, designed to evaluate the LVLMs' ability to leverage domain-specific knowledge to identify object categories in images across different levels of granularity within hierarchical taxonomies.

***Machine-oriented Evaluation***:

- *Image Retrieval*: retrieves images belonging to multiple subordinate categories of a meta-category by measuring the similarity of their visual features.
- *Image Classification*: recognizes images into fine-grained categories, either within a single meta-category (*e.g.*, species of animals, models of cars) or across multiple meta-categories, assessing the model's ability to handle diverse data sources and make accurate predictions.

More evaluation tasks details are presented in Appendix A.1.

## 3.2    DATA CURATION

To ensure the quality and diversity of the data, we source images for the `FG-BMK` benchmark from 12 well-established fine-grained datasets, which helps avoid issues of inconsistent image quality and the risk of mislabeling often found in web-sourced images (Radford et al., 2021).

- *Attribute Recognition*: We design true/false and multiple-choice questions based on the fine-grained annotations of the images. For the multiple-choice questions, the options include all possible attribute candidates. In contrast, for the true/false questions, half are negative samples, pairing the image with an incorrect attribute, and the other half are positive samples.
- *Knowledge Bias Estimation*: In addition to positive samples, for each fine-grained category label, we pair it with images from other subcategories within the same super-category to generate negative samples for true/false questions.
- *Hierarchical Granularity Recognition*: We design true/false, multiple-choice, and short-answer questions at various levels of granularity based on the hierarchical taxonomy labels of the images. For true/false questions, negative samples are created by pairing images with incorrect labels at the same hierarchical level (*e.g.*, an image of Aves (birds) paired with Insecta (insects)). For multiple-choice questions, options are drawn from different categories within the same parent category of the hierarchical taxonomy (*e.g.*, species-level options such as black-footed albatross and Laysan albatross, both within the genus Albatross). For short-answer questions, the model is prompted to generate its response directly.
- *Image Retrieval and Classification*: We use the original labels directly from the datasets. For classification across meta-categories, we combine subcategories from different meta-categories into a unified training/testing set, where all fine-grained categories are mixed for training.

More specifically, for each task in the human-oriented evaluation, we manually design several question templates to ensure both diversity and comprehensive coverage. More details of curated data can be found in Appendix A.2.

## 4    OBSERVATIONS AND DISCUSSIONS

### 4.1    MODELS UNDER EVALUATION

Given the diverse landscape of existing LVLMs, we select nine widely-used open-source LVLMs, two closed-source models (GPT-4o-1120 (OpenAI, 2023) and Gemini-2.0-flash (Gemini Team, 2024)) and one purely visual model with varying training strategies, as shown in Table 1. To enable clear comparisons and isolate the factors influencing performance, we use earlier versions of models from each family in machine-oriented evaluation, where their distinct characteristics are more evident and less affected by additional tricks or complex modifications. Further details about the evaluated models can be found in Appendix B.

Table 1: Training strategies of the open-source evaluated models. "DINOv2" is a purely visual model. "Con" denotes contrastive loss, "Gen" generative loss, "Mat" image-text matching loss, "Rec" reconstruction loss used in BEiT3, and "Dis" distillation loss used in DINOv2.

| Model | Vision Size | Loss Function | | | | | Training Data | | |
|---|---|---|---|---|---|---|---|---|---|
| | | Con | Gen | Mat | Rec | Dis | < 0.1B | 0.1B ∼ 1B | > 1B |
| InternVL3-7B (Zhu et al., 2025) | ViT-L | ✓ | ✓ | ✓ | | ✓ | | | ✓ |
| InternVL-Chat (Chen et al., 2024) | ViT-6B | ✓ | ✓ | ✓ | | | | | ✓ |
| LLaVA-1.5-7B (Liu et al., 2024a) | ViT-L | | ✓ | | | | ✓ | | |
| Qwen2.5-VL-7B (Bai et al., 2025) | ViT-600M | ✓ | ✓ | ✓ | | ✓ | | | ✓ |
| Qwen-VL-Chat (Bai et al., 2023) | ViT-G | | ✓ | | | | | | ✓ |
| BLIP-2-XL (Li et al., 2023) | ViT-G | ✓ | ✓ | ✓ | | | | ✓ | |
| EVA-CLIP (Sun et al., 2023) | ViT-L | ✓ | | | | | | | ✓ |
| BEiT3 (Wang et al., 2023) | ViT-L | | | | ✓ | | ✓ | | |
| CoCa (Yu et al., 2022) | ViT-L | ✓ | ✓ | | | | | | ✓ |
| DINOv2 (Oquab et al., 2023) | ViT-L | ✓ | | | | ✓ | | ✓ | |

## 4.2 Human-oriented Evaluation

In the human-oriented evaluation, we assess the ability of LVLMs to understand visual content and utilize domain-specific knowledge through conversational interactions. Specifically, we first evaluate the domain-specific knowledge embedded in LVLMs by measuring their accuracy in identifying object categories across different levels of granularity, as illustrated in Figure 2. Then, we investigate whether LVLMs exhibit knowledge biases when recognizing different fine-grained categories by ranking their accuracy in answering true/false questions for each category, cf. Figure 3. Additionally, we examine the capability of LVLMs to recognize fine-grained attributes, with results summarized in Table 2. Lastly, Table 3 compares the classification accuracy of LVLMs against state-of-the-art fine-grained tailored models on datasets spanning various domains. Based on these observations, we draw the following conclusions.

**LVLMs struggle to distinguish excessively fine-grained categories.** As shown in Figure 2, taking InternVL3 (Zhu et al., 2025) as an example, its accuracy in answering true/false and multiple-choice questions declines as the granularity becomes finer. At the class level (*e.g.*, *"Is the class of the object in this image an Insecta/Aves?"*), the model achieves 99.76% accuracy on multiple-choice questions and 99.77% on true/false questions.[1] However, as the granularity narrows to the genus level, where negative categories are drawn from different genera within the same class (*e.g.*, *"Is the object in this image an albatross or a gull?"*), the model's accuracy on multiple-choice questions drops to 90.75%, a decrease of 9.01%. When the granularity is further refined to the species level, where negative categories are from different

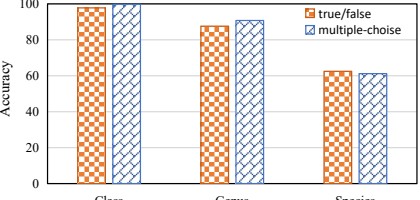

Figure 2: Results of InternVL3 (Zhu et al., 2025) on true/false and multiple-choice questions across different levels of granularity on the *CUB-200-2011* (Wah et al., 2011) dataset. The $x$-axis denotes the granularity of the recognition questions.

species within the same genus (*e.g.*, *"Is the object in this image a black-footed albatross/Laysan albatross?"*), the accuracy drops further to 62.48% on true/false questions and 61.18% on multiple-choice questions. This demonstrates that the model struggles to distinguish between closely related species. The results for other LVLMs exhibit similar trends to those for InternVL3. Additional examples of multiple-choice and true/false questions can be found in Appendix C.1.

**The inconsistent recognition accuracy of LVLMs across fine-grained categories can be attributed to the characteristics of their training data and the underlying LLM base.** To examine whether LVLMs exhibit knowledge bias in recognizing different fine-grained categories, we ranked their accuracy in answering true/false questions for each fine-grained category. As shown in Figure 3, using LLaVA (Liu et al., 2024a) as an example, the model demonstrates highly inconsistent recognition abilities across categories, achieving nearly 90% accuracy for some while dropping to approximately 30% for others.

---

[1] When questions are relatively simple, LVLMs achieve very high accuracy. The higher accuracy on multiple-choice questions compared to true/false questions may result from randomness.

Table 2: Attribute recognition accuracy of InternVL3 (Zhu et al., 2025) on the *CUB-200-2011* (Wah et al., 2011) dataset (values in parentheses represent the average accuracy for each attribute).

| Color Attribute (47.40) | | | | | | | |
|---|---|---|---|---|---|---|---|
| belly color | 58.49 | back color | 34.98 | bill color | 51.31 | breast color | 54.25 |
| crown color | 55.30 | eye color | 84.59 | forehead color | 53.32 | leg color | 44.01 |
| nape color | 39.24 | throat color | 52.77 | under tail color | 34.69 | underparts color | 56.20 |
| upper tail color | 37.30 | upperparts color | 28.75 | wing color | 30.16 | primary color | 43.05 |
| Pattern Attribute (50.13) | | | | | | | |
| back pattern | 40.94 | belly pattern | 68.13 | breast pattern | 65.12 | head pattern | 35.92 |
| tail pattern | 41.64 | wing pattern | 49.04 | | | | |
| Shape Attribute (30.95) | | | | | | | |
| bill shape | 37.61 | shape | 52.37 | tail shape | 10.42 | wing shape | 23.39 |
| Length Attribute (71.03) | | | | Size Attribute (52.55) | | | |
| bill length | | | 71.03 | size | | | 52.55 |

We hypothesize that this inconsistency arises either from the uneven representation of fine-grained knowledge in the training data or from the inherent difficulty LVLMs face in learning certain fine-grained categories. To test this hypothesis, we fine-tuned LVLMs on datasets with balanced occurrences of fine-grained categories and evaluated their performance. As indicated by the yellow dots in Figure 3, the fine-tuned LLaVA shows consistently strong recognition ability across all fine-grained categories. It suggests that the knowledge bias is primarily caused by the uneven representation of fine-grained knowledge in training data, rather than the inherent difficulty of learning specific categories.

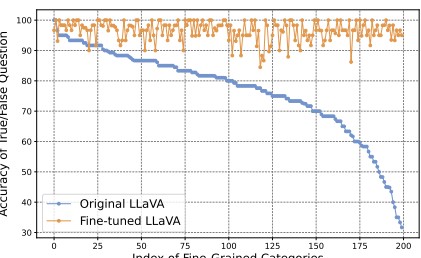

Figure 3: Comparison of the original (blue dots) and fine-tuned (yellow dots) LLaVA models on occurrence-balanced fine-grained bird categories. True/false accuracy per category is ranked.

To further validate this hypothesis, we examined the occurrence frequency of fine-grained categories in the LVLM's training data. Interestingly, we found that such categories are almost absent from the training data. This indicates that the observed inconsistency in recognition ability is inherited from the language model underlying the LVLM, rather than being solely a consequence of the visual model itself. Additional results for other LVLMs exhibit similar trends and can be found in Appendix C.2.

**LVLMs exhibit significant room for improvement in recognizing fine-grained attributes.** As shown in Table 2 of the paper and Table 14 in the appendix, LVLMs exhibit relatively stronger recognition abilities for attributes like pattern, size and length compared to others. For example, InternVL3 (Zhu et al., 2025) and Qwen2.5-VL (Bai et al., 2025) achieve 50.13% and 45.12% average accuracy for pattern recognition, but only 30.95% and 29.30% for shape recognition. Notably, only a few attributes achieved an accuracy above 70%, and several attributes scored as low as 10%, highlighting significant room for improvement in LVLMs' fine-grained attribute recognition.

We also observe model-specific differences in attribute recognition. For instance, while InternVL3 struggles more with pattern recognition compared to size, Gemini-2.0-flash (Gemini Team, 2024) shows the opposite trend. Additionally, we find that the latest LVLMs show significant improvements in pattern and length recognition, while their advancements in color and shape recognition are more limited. Detailed results of the attribute recognition task are provided in Appendix C.3.

**LVLMs do not outperform fine-grained tailored models in fine-grained tasks.** In Table 3, we compare the recognition accuracy of LVLMs with that of fine-grained tailored models. While LVLMs demonstrate notable performance in fine-grained recognition, their accuracy—whether in short-answer questions or using the linear classifier method—falls short of that achieved by fine-grained tailored models. This disparity may stem from the fact that LVLMs are primarily optimized for general tasks, with less emphasis placed on fine-grained domain capabilities (*e.g.*, CAP (Behera et al., 2021) uses context-aware attentional pooling to capture hierarchical contextual information from pixel to region to image level, leading to a 1.53% improvement in classification accuracy).

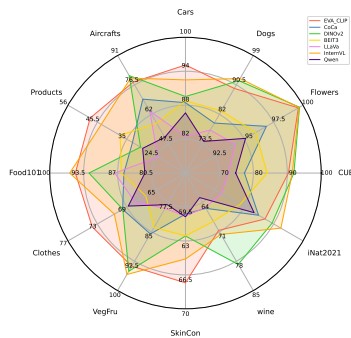

Figure 4: Retrieval results of LVLM visual features on twelve fine-grained datasets. Different colors represent different models.

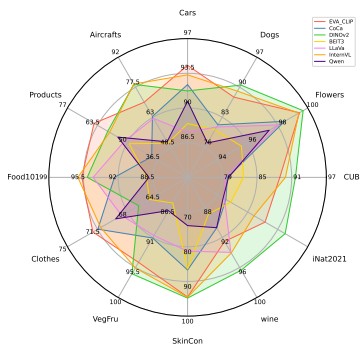

Figure 5: Classification results of LVLM visual features on twelve fine-grained datasets. Different colors represent different models.

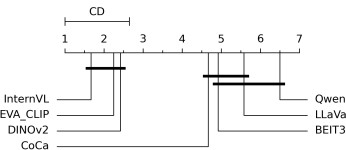

Figure 6: Nemenyi statistical test results for fine-grained retrieval. Black horizontal lines indicate the critical distance (CD), grouping models with no significant performance differences.

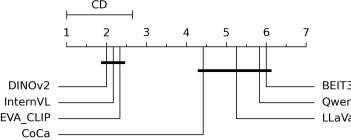

Figure 7: Nemenyi statistical test results for fine-grained recognition. Black horizontal lines indicate the critical distance (CD), grouping models with no significant performance differences.

Due to the architectural paradigm of LVLMs (typically ViT + MLP + LLM), these components cannot be directly applied. However, the core idea of attending to hierarchical-level details can still be incorporated into LVLMs (*e.g.*, InternVL3 (Zhu et al., 2025) not only takes the full image as input, but also divides the image into parts to obtain more fine-grained visual representations). Enhancing LVLMs performance in specialized areas (*e.g.*, fine-grained recognition) while preserving their strengths in general tasks presents an important and promising direction for future research.

Table 3: Comparison of LVLMs and fine-grained tailored models on classification tasks. "SA" denotes LVLM results fine-tuned on fine-grained datasets for short-answer questions, "LC" represents linear classifier results using LVLM visual features, and "FG-Tailored" refers to state-of-the-art fine-grained tailored model results.

| Datasets | SA | LC | FG-Tailored |
|---|---|---|---|
| *CUB-200-2011* | 85.60 | 91.65 | 93.10 (Diao et al., 2022) |
| *Stanford Dog* | 86.49 | 90.50 | 97.30 (Bera et al., 2022) |
| *Stanford Car* | 90.55 | 94.30 | 97.10 (Liu, 2024) |
| *Food-101* | 95.25 | 95.67 | 98.60 (Behera et al., 2021) |
| *FGVC Aircraft* | 66.19 | 78.88 | 95.40 (Sikdar et al., 2024) |

### 4.3 MACHINE-ORIENTED EVALUATION

Compared to human-oriented evaluation, machine-oriented evaluation takes a more direct approach to assess the visual feature representations of LVLMs by employing two fundamental visual tasks: image retrieval and image recognition. This evaluation focuses on two key aspects: 1) Discriminability—the ability of visual features to distinguish fine-grained categories, and 2) Robustness—maintaining accuracy under feature perturbations.

To evaluate discriminability, we start by analyzing the performance of visual features on retrieval and classification tasks, following the setting of DINOv2 (Oquab et al., 2023). Figures 4 and 5 display retrieval and classification results across twelve fine-grained datasets, while Figures 6 and 7 provide the Friedman test results for these tasks.

We then increase classification difficulty by merging fine-grained categories from different super-categories into a unified training and testing set. Figure 8 presents classification accuracy within a single super-category and across multiple meta-categories. Visual features with strong discriminability should maintain high accuracy even with diverse data sources. Additionally, we analyze how the vision encoder size in LVLMs impacts classification accuracy, as shown in Figure 9.

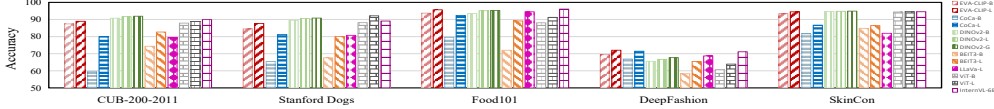

Figure 8: Classification results of LVLM visual features on fine-grained datasets. "Single" denotes accuracy from training on a single meta-category, while "Multiple" reflects accuracy from training on a unified dataset combining multiple meta-categories.

Figure 9: Classification results with different vision encoder sizes. Bars filled with different patterns represent different models, with darker patterns indicating larger vision encoder sizes.

Furthermore, we examine how vision-language alignment—a critical component of LVLMs—affects visual feature quality. Specifically, we compare classification accuracy between original visual features and those aligned to textual features in fine-grained classification tasks, as shown in Table 4.

To investigate robustness, we introduce perturbations to visual features using projected gradient descent (Madry et al., 2018) and analyze their impact on classification accuracy. Results of this experiment are presented in Table 5.

**The contrastive training paradigm in LVLMs effectively enhances the fine-grained discriminability of visual features.** In Figure 4 and Figure 5, vision encoders trained with contrastive paradigms (*e.g.*, EVA-CLIP, InternVL, DINOv2) outperform those trained with reconstruction (BEiT3) and generative paradigms (Qwen) on fine-grained retrieval and classification tasks. As shown in Figure 6 and Figure 7, Nemenyi test results also indicate that InternVL, EVA-CLIP, and DINOv2 perform significantly better than Qwen and BEiT3. In multi meta-category classification (cf. Figure 8), EVA-CLIP maintains high accuracy, with an average drop of just 1.96% compared to the single-category setting, while Qwen and BEiT3 show larger drops of 4.16% and 7.41%, respectively. This underscores the effectiveness of contrastive paradigms in fine-grained tasks.

In experiments of vision encoder size, as shown in Figure 9, we observe that DINOv2-B, with a smaller vision encoder, achieves higher classification accuracy compared to BEiT3-L, with a larger vision encoder, outperforming by 8.08% on *CUB-200-2011* and 9.49% on *Stanford Dogs*. We attribute these findings to the limitations of generative and reconstruction training paradigms, which fail to adequately distinguish between features of different fine-grained categories, while maintaining similarity within the same. This limitation hinders their performance on fine-grained tasks.

**The alignment strategy in LVLMs might impair the fine-grained discriminability of visual features.** Alignment plays a critical role in LVLMs by bridging the gap between visual and textual features. To assess its impact on fine-grained visual feature discriminability, we compare the classification accuracy of LLaVA's (Liu et al., 2024a) original visual features with those aligned to textual features on fine-grained classification tasks. As shown in the first two columns of Table 4, the original features demonstrate superior classification

Table 4: Classification accuracy of LLaVA visual features before and after alignment. "Origin" denotes original features from the vision encoder. "Aligned" denotes features aligned to text with inconsistent granularity, while "Aligned-FG" denotes those aligned to fine-grained text.

| Datasets | Origin | Aligned | Aligned-FG |
|---|---|---|---|
| *CUB-200-2011* | 79.77 | 73.17 | 75.06 |
| *Stanford Dogs* | 81.24 | 78.14 | 80.69 |
| *Stanford Cars* | 87.57 | 83.90 | 85.63 |
| *Food-101* | 94.27 | 93.35 | 94.32 |
| *DeepFashion* | 69.94 | 67.30 | 67.75 |

performance, outperforming the aligned ones by an average of 3.39%.

This decline in performance can be attributed to two key factors. First, aligning visual and textual features may introduce distortions due to inconsistencies between their respective feature spaces. Second, granularity inconsistencies in LVLMs' alignment data—where fine-grained objects in images

Table 5: Classification results of LVLMs' original and perturbed visual features on the fine-grained dataset *CUB-200-2011* and the generic dataset *CIFAR-100*. "Origin" refers to results with original features, while "Perturbed" indicates results with perturbed features.

| Datasets | EVA-CLIP | | CoCa | | DINOv2 | | ViT | |
|---|---|---|---|---|---|---|---|---|
| | Origin | Perturbed | Origin | Perturbed | Origin | Perturbed | Origin | Perturbed |
| *CIFAR-100* | 93.05 | 50.76 | 86.94 | 52.23 | 93.38 | 42.39 | 89.81 | 72.15 |
| *CUB-200-2011* | 88.95 | 24.94 | 79.89 | 23.40 | 91.64 | 25.94 | 88.83 | 73.85 |

are paired with coarse-grained textual descriptions (as demonstrated in our qualitative analysis in Appendix D.1)—negatively affect the discriminability of the aligned visual features.

To validate the impact of granularity inconsistency, we construct a new alignment dataset where the textual descriptions match the granularity of the fine-grained objects in images. We then retrain the alignment module in LLaVA using this dataset. As shown in Table 4, the classification accuracy of visual features aligned with fine-grained textual content improves significantly, with an increase of 2.55% on *Stanford Dogs* and 1.73% on *Stanford Cars*. While the aligned visual features still underperform compared to the original features, these results underscore the detrimental effects of granularity inconsistency on visual feature discriminability and highlight the importance of ensuring granularity alignment between image and textual content. Further experiments in Appendix D.2 reveal that enhancing the fine-grained discriminability of visual features during the alignment stage can lead to improved LVLM performance on fine-grained tasks.

**Larger vision encoder size or larger scale of web data provides limited improvement in the fine-grained discriminability of visual features.** Regarding vision encoder size, as shown in Figure 9, increasing the size of DINOv2's vision encoder from DINOv2-B to DINOv2-L results in only a 0.6% improvement in average classification accuracy, while further increasing from DINOv2-L to DINOv2-G leads to just a 0.3% improvement. The classification accuracy of visual features in InternVL-6B is not superior to that of DINOv2-L, suggesting that merely enlarging the vision encoder has limited impact on enhancing the fine-grained discriminability of visual features.

Regarding the amount of training data, as shown in Figure 7, the vision encoder of EVA-CLIP, trained on over 2 billion data samples, does not outperform DINOv2, which was trained on 142 million samples, in fine-grained classification and retrieval tasks. We attribute this difference to the quality of the training data, that DINOv2's dataset is carefully curated from a large pool of data, whereas EVA-CLIP relies on crawled web data. A similar trend is observed when comparing DINOv2 with InternVL (6B samples), suggesting that simply increasing the scale of training data, without considering its quality, offers limited improvement in fine-grained discriminability of visual features.

**Visual features in LVLMs are more susceptible to perturbations in fine-grained tasks.** As shown in Table 5, perturbing the visual features of EVA-CLIP causes a significant drop in classification accuracy on the fine-grained dataset *CUB-200-2011* (from 88.95% to 24.94%). In contrast, on the generic dataset *CIFAR-100* (Krizhevsky & Hinton, 2009), the accuracy declines more modestly (from 93.05% to 50.76%). Similar trends are observed for CoCa and DINOv2, indicating that fine-grained tasks are more susceptible to perturbations than generic tasks. We attribute this vulnerability to coarse-grained, noisy training data, which limits feature discriminability across fine-grained categories, making it hard to distinguish them when visual features are perturbed.

In contrast, the Vision Transformer (Wightman, 2019) trained on the *ImageNet* (Deng et al., 2009) dataset with cross-entropy loss demonstrates superior robustness to perturbations. It shows only minor declines in classification accuracy on both fine-grained and generic datasets. This suggests that adopting alternative training paradigms or incorporating high-quality, fine-grained data (as seen in *ImageNet*) during training could help improve the robustness of visual features in LVLMs.

## 5 CONCLUDING REMARKS

We conducted a comprehensive evaluation to investigate the capabilities of LVLMs on fine-grained visual tasks, leading to several key findings. First, we observed that the contrastive training paradigm significantly enhances the fine-grained discriminability of visual features, whereas generative and reconstruction-based paradigms tend to underperform in this aspect. Second, aligning visual features

with textual features can impair their fine-grained discriminability, particularly when there is a mismatch in granularity between the paired image-text data. Third, our experiments revealed that LVLMs and LVMs are more susceptible to feature perturbations in fine-grained tasks compared to generic vision tasks, highlighting a vulnerability that merits further attention. Fourth, while LVLMs demonstrate strong capabilities in visual appearance perception, they face notable challenges in fine-grained category reasoning, which depends heavily on recognizing subtle visual attributes. Finally, despite their advancements, LVLMs still lag behind specialized fine-grained models, underscoring the need for task-specific enhancements. These findings highlight key limitations of LVLMs, identifying critical areas for improvement, such as training paradigms, robustness to perturbations, and reasoning capabilities. Our work lays the groundwork for future research to advance LVLM performance on fine-grained visual tasks.

**Ethics Statement**   All data used in this study are collected from publicly available datasets that comply with standard research ethics. The benchmark tasks and annotations in `FG-BMK` are constructed through automatic or rule-based question generation without involving human subjects. Although the dataset SkinCon relate to medical images, it is fully anonymized and released under research-permissive licenses. Our use strictly follows its terms and involves no patient-identifying information, clinical decision-making, or sensitive personal data. We do not release any private or sensitive information, nor do our methods introduce additional safety, fairness, or legal concerns. Overall, FG-BMK provides a practical and ethically compliant testbed for evaluating LVLMs in fine-grained real-world scenarios.

**Reproducibility Statement**   We ensure reproducibility by using only publicly available datasets and by clearly describing our benchmark construction process in Section 3 and Appendix A. The evaluation pipeline, including prompt design, scoring methods, and model configuration, is detailed in Section 3, Appendix A and Appendix B. We provide extensive implementation details and dataset statistics in the Appendix. All code, data splits, and evaluation scripts has be released.

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

## A   THE EVALUATION BENCHMARK

### A.1   EVALUATION TASK DETAILS

In Section 3.1, we have described each evaluation task. Here, we provide further details. In the Knowledge Bias Estimation task, to uncover potential knowledge biases across different fine-grained categories, we pair each image with its corresponding fine-grained label to generate positive samples for true/false questions. For constructing negative samples, each image is paired with a single fine-grained label randomly selected from other subcategories within the same super-category. For each fine-grained category, we calculate the LVLM's accuracy on all coresponding true/false questions as a measure of its understanding of that category's knowledge.

In the cross meta-class classification task, we follow the DINOv2 (Oquab et al., 2023) method to train the model on a unified training set where fine-grained categories from different datasets are combined. The model is then tested on each individual dataset to evaluate its performance.

### A.2   DATA CURATION

**Dataset**   We source images for the `FG-BMK` benchmark from 12 fine-grained datasets. These datasets cover a wide range of meta-classes, with different categories and sample, providing a comprehensive assessment of LVLMs capabilities on fine-grained tasks across different domains. Table 6 indicates their meta-classes, the amount of samples, the number of categories. For all datasets, we construct human-oriented evaluation questions based on their test sets. We use the original labels directly from the datasets for the machine-oriented evaluation.

Table 6: Details of 12 fine-grained datasets sorted by their numbers of categories. "Meta-class" refers to a high-level categorization of the dataset. "Categories" refers to the number of fine-grained categories. "Samples" refers to the total number of samples in each dataset.

| Datasets | Meta-class | Categories | Samples |
|---|---|---|---|
| *Wine (Tianchi, 2021)* | Industrial | 11 | 4,516 |
| *DeepFashion (Liu et al., 2016)* | Clothes | 46 | 18,000 |
| *SkinCon (Daneshjou et al., 2022)* | Dermatology | 48 | 3,866 |
| *Flowers102 (Nilsback & Zisserman, 2008)* | Flower | 102 | 7,169 |
| *Food101 (Bossard et al., 2014)* | Food | 101 | 101,000 |
| *FGVC Aircraft (Maji et al., 2013)* | Aircraft | 100 | 6,667 |
| *Stanford Dogs (Khosla et al., 2011)* | Dog | 120 | 20,580 |
| *Stanford Cars (Krause et al., 2013)* | Car | 196 | 16,185 |
| *CUB-200-2011 (Wah et al., 2011)* | Bird | 200 | 11,788 |
| *VegFru (Hou et al., 2017)* | Vegetable | 292 | 146,131 |
| *Products-10K (Bai et al., 2020)* | Retail | 9,691 | 197,307 |
| *iNat2021 (Van Horn et al., 2021)* | Plants | 10,000 | 2,786,843 |

**Human-oriented Question Templates**   When constructing true/false, multiple-choice, short answer questions for each task in human-oriented evaluation, we manually design several question templates to ensure both diversity and comprehensive coverage. Figure 10 illustrates the question templates we use for generating the tasks.

We also expanded the original template set to 10 diverse human-written prompts and reconstructed the multiple-choice questions in the human-oriented benchmark to examine the potential impact of linguistic diversity. As shown in Table 7 and Table 8, increasing the number of templates leads to only minor changes in accuracy, and the overall LVLM behavior and observed trends remain consistent. Therefore, as long as the template clearly states the question, the effect of the template quantity on the results is negligible.

**Question Template for Attribute Recognition Task**

**True/False Question:**
Is the wing color of the bird {color}?
Is the breast pattern of the bird {pattern}?

**Multiple-choice Question:**
What is the wing color of the bird? Choose one from the following list: {options}.

**Question Template for Knowledge Bias Estimation Task**

**True/False Question:**
Is the species of the {meta_class} {fine_grained_category}? Answer with yes or no.
Does this {meta_class} belong to the species known as {fine_grained_category}? Answer with yes or no.
Is the {meta_class} species in this photo a {fine_grained_category}? Answer with yes or no.
From your observation, is the species of the {meta_class} shown a {fine_grained_category}? Answer with yes or no

**Question Template for Hierarchical Granularity Recognition Task**

**True/False Question:**
Is the species of the {meta_class} {fine_grained_category}? Answer with yes or no.
Does this {meta_class} belong to the species known as {fine_grained_category}? Answer with yes or no.
Is the {meta_class} species in this photo a {fine_grained_category}? Answer with yes or no.
From your observation, is the species of the {meta_class} shown a {fine_grained_category}? Answer with yes or no

**Multiple-choice Question:**
What is the species of the {meta_class} in this image? Choose one answer from the following list: {options}.
Answer the question using a single word or phrase.
Can you identify the species of this {meta_class}? Choose one answer from the following list: {options}.
Answer the question using a single word or phrase.
Which species does this {meta_class} in the photo belong to? Choose one answer from the following list: {options}. Answer the question using a single word or phrase.
Observing the {meta_class} in the image, which of the following species is it? Choose one answer from the following list: {options}. Answer the question using a single word or phrase.

**Short Answer question:**
What species of the {meta_class} is shown in the image? Directly answer with species names.
Can you identify the species of this {meta_class} from the image? Directly answer with species names.
Which species does the {meta_class} in this photo belong to? Directly answer with species names.
Based on your observation, what species of the {meta_class} is depicted? Directly answer with species names.

Figure 10: Question templates for each task in huamn-oriented evaluation.

Table 7: Attribute recognition accuracy of InternVL3 (Zhu et al., 2025) using original and extended prompts on the *CUB-200-2011* (Wah et al., 2011) dataset (values in parentheses represent the average accuracy for each attribute). Accuracy are shown in the format "original / extended", with the left representing accuracy using the original prompt and the right using the extended prompt.

| Color Attribute (47.40 / 47.45) | | | | | |
|---|---|---|---|---|---|
| belly color | 58.49 / 60.04 | back color | 34.98 / 36.33 | bill color | 51.31 / 49.64 |
| breast color | 54.25 / 55.91 | crown color | 55.30 / 54.01 | eye color | 84.59 / 82.96 |
| forehead color | 53.32 / 51.90 | leg color | 44.01 / 45.67 | nape color | 39.24 / 38.02 |
| throat color | 52.77 / 54.53 | under tail color | 34.69 / 35.80 | underparts color | 56.20 / 55.08 |
| upper tail color | 37.30 / 38.77 | upperparts color | 28.75 / 27.50 | wing color | 30.16 / 31.88 |
| primary color | 43.05 / 41.29 | | | | |
| **Pattern Attribute (50.13 / 50.28)** | | | | | |
| back pattern | 40.94 / 39.38 | belly pattern | 68.13 / 67.00 | breast pattern | 65.12 / 66.87 |
| head pattern | 35.92 / 34.66 | tail pattern | 41.64 / 42.93 | wing pattern | 49.04 / 50.84 |
| **Shape Attribute (30.95 / 31.01)** | | | | | |
| bill shape | 37.61 / 36.41 | shape | 52.37 / 50.60 | tail shape | 10.42 / 12.04 |
| wing shape | 23.39 / 24.98 | | | | |
| **Length Attribute (71.03 / 69.71)** | | | Size Attribute (52.55 / 54.21) | | |
| bill length | | 71.03 / 69.71 | size | | 52.55 / 54.21 |

Table 8: Results of InternVL3 using original and extended prompts on true/false (TF) and multiple-choice (MC) questions across different levels of granularity on the *CUB-200-2011* dataset. Results are shown in the format "original / extended".

| TF | | | | | |
|---|---|---|---|---|---|
| Class | 98.79 / 98.03 | Genus | 85.69 / 86.19 | Species | 61.88 / 62.34 |
| **MC** | | | | | |
| Class | 99.42 / 99.58 | Genus | 88.13 / 87.62 | Species | 60.15 / 59.23 |

## B  EVALUATED MODELS

As shown in Table 9, we select nine widely-used open-source LVLMs, two closed-source models (GPT-4o (OpenAI, 2023) and Gemini-2.0-flash (Gemini Team, 2024)) and one purely visual model, each of which employs a distinctive training recipes, including variations in vision encoder, language model, training losses and data.

Table 9: Configurations of the evaluated models. "DINOv2-L" is a purely visual model. "Con" stands for the contrastive loss, "Gen" for the generative loss, "Mat" for the image-text matching loss, "Rec" for the reconstruction loss as used in BEiT3 (Wang et al., 2023), and "Dis" for the distillation loss as applied in DINOv2 (Oquab et al., 2023).

| Model | Component | | Loss Function | | | | |
|---|---|---|---|---|---|---|---|
| | Vision Model | Language Model | Con | Gen | Mat | Rec | Dis |
| InternV3-7B | InternViT-L | Qwen2.5-7B | ✓ | ✓ | ✓ | | ✓ |
| InternVL-Chat-V1.1 | InternViT-6B | LLaMA2-13B | ✓ | ✓ | ✓ | | |
| LLaVA-1.5-7B | CLIP-L | Vicuna-7B | | ✓ | | | |
| Qwen2.5-VL | CLIP-600M | Qwen2.5-7B | ✓ | ✓ | ✓ | | ✓ |
| Qwen-VL | Openclip-G | Qwen-7B | | ✓ | | | |
| BLIP-2-FLAN-T5-XL | EVA-CLIP-G | FlanT5-XL | ✓ | ✓ | ✓ | | |
| EVA02-CLIP-L | EVA02-L | CLIP-L | ✓ | | | | |
| BEiT3-L-ITC | CLIP-L | CLIP-L | | | | ✓ | |
| CoCa-L | CLIP-L | CLIP-L | ✓ | ✓ | | | |
| DINOv2-L | CLIP-L | / | ✓ | | | | ✓ |

- **EVA-CLIP** (Sun et al., 2023) aligns visual and textual features using contrastive loss, leveraging over 2 billion web image-text pairs and advanced optimization techniques.

- **InternVL3** (Zhu et al., 2025) adopts a unified pre-training approach over both multimodal and pure-text data, enhanced by variable visual position encoding (V2PE) and advanced post-training strategies for improved scalability and effectiveness.
- **InternVL** (Chen et al., 2024) leverages contrastive, matching, and generative losses in a multi-stage training process, with a large-scale vision encoder and over 6 billion image-text pairs to align visual and textual representation.
- **BLIP-2** (Li et al., 2023) bridges the modality gap between frozen image encoders and LLMs using a lightweight Q-Former, leveraging contrastive, matching, and generative loss in a two-stage pre-training process over 129 million data with fewer trainable parameters.
- **Qwen2.5-VL** (Bai et al., 2025) combines dynamic-resolution Vision Transformer with Window Attention to reduce computational cost while preserving native image resolution.
- **Qwen-VL** (Bai et al., 2023) employs a three-stage training process with generative loss, using a VL adapter to align visual and textual features while reducing computational cost over 1.4 billion image-text pairs.
- **CoCa** (Yu et al., 2022) adopts task-specific attentional pooling to tailor visual representations for different training objectives, applying contrastive loss to train the first half of the decoder and generative loss to train the full decoder in an end-to-end manner over 5 billion image-text pairs.
- **BEIT3** (Wang et al., 2023) treats images as a foreign language, leveraging a mask-then-predict objective over 36 million image-text pairs to unify vision and language pretraining, and introduces a multiway transformer architecture for general-purpose modeling.
- **LLaVA** (Liu et al., 2024a) aligns visual and textual features using a simple MLP with generative loss, leveraging 1.2 million GPT-4 (OpenAI, 2023) generated multimodal instruction-following data for training.
- **DINOv2** (Oquab et al., 2023) uses a self-supervised learning approach, leveraging knowledge distillation and a mask-then-predict strategy over 142 million images to train the vision encoder.

For all our evaluated model, we follow their official configurations to run the inference. We set the temperature of all open-source models to 0, while keeping the default for closed-source APIs.

## C HUMAN-ORIENTED EVALUATIONS

### C.1 RESULTS OF HIERARCHICAL GRANULARITY RECOGNITION

Figure 2 shows InternVL3's (Zhu et al., 2025) accuracy in answering true/false and multiple-choice questions within hierarchical granularity recognition task on *CUB-200-2011* dataset. In Figure 11, we present additional results for GPT-4o (OpenAI, 2023), Gemini-2.0-flash (Gemini Team, 2024), Qwen2.5-VL (Bai et al., 2025), LLaVA (Liu et al., 2024a) and InternVL (Chen et al., 2024) on *CUB-200-2011* (Wah et al., 2011) and *iNat2021* (Van Horn et al., 2021) datasets. As shown in the experiments, the accuracy of all models decreases as the granularity becomes finer. When the granularity level reaches the finest level, the models struggle to distinguish between closely related species.

### C.2 RESULTS OF KNOWLEDGE BIAS ESTIMATION

In Figure 3, we observe that LLaVA exhibit highly inconsistent recognition abilities across categories. We also conduct experiments with Qwen2.5-VL, GPT-4o, and Gemini-2.0-flash on fine-grained datasets such as Aircraft (Maji et al., 2013), Flowers102 (Nilsback & Zisserman, 2008) and Stanford Dogs (Khosla et al., 2011). As shown in Figure 12 and Figure 13, all LVLMs display similar trends, indicating inconsistent recognition abilities across fine-grained categories. However, after fine-tuned on datasets with balanced occurrences of fine-grained categories, LVLMs demonstrate remarkable recognition abilities across all fine-grained categories.

To construct datasets with balanced occurrences of fine-grained categories, we select an equal number of images from each category. Then we generate the same number of true/false questions for each fine-grained category, thereby fine-tuning the LVLMs in a way that each category receives balanced representation.

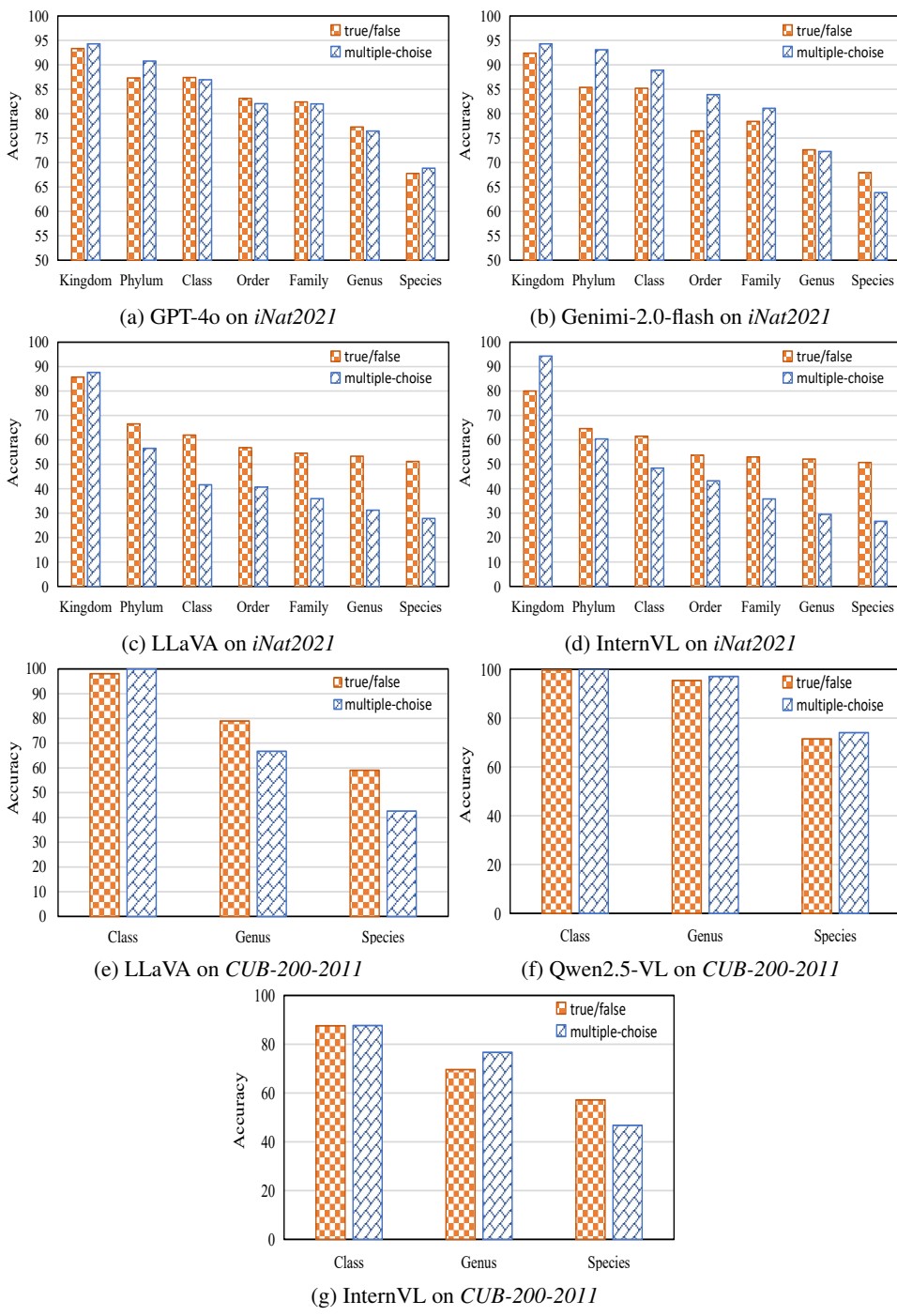

Figure 11: Results of GPT-4o (OpenAI, 2023), Gemini-2.0-flash (Gemini Team, 2024), Qwen2.5-VL (Bai et al., 2025), LLaVA (Liu et al., 2024a) and InternVL (Chen et al., 2024) on true/false and multiple-choice questions across different levels of granularity on *CUB-200-2011* (Wah et al., 2011) and *iNat2021* (Van Horn et al., 2021) dataset. The x-axis denotes the granularity of the recognition questions.

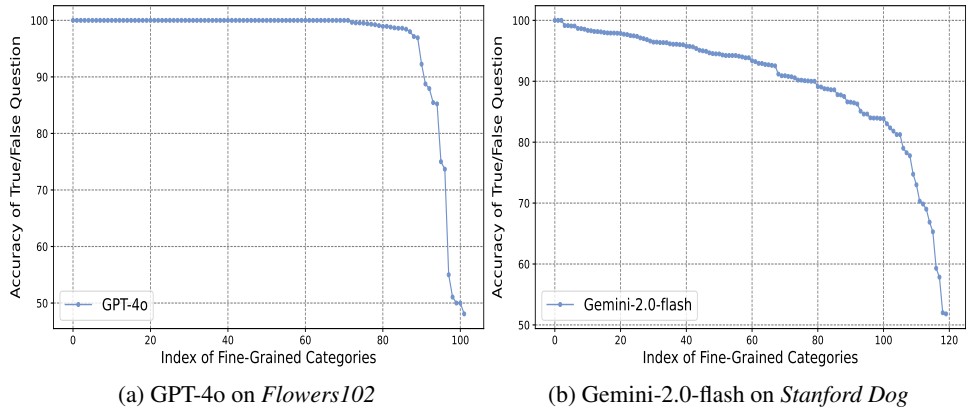

(a) GPT-4o on *Flowers102*  (b) Gemini-2.0-flash on *Stanford Dog*

Figure 12: Knowledge bias estimation results of two closed-source models. True/false question accuracy for each category is ranked, with blue dots representing the original model.

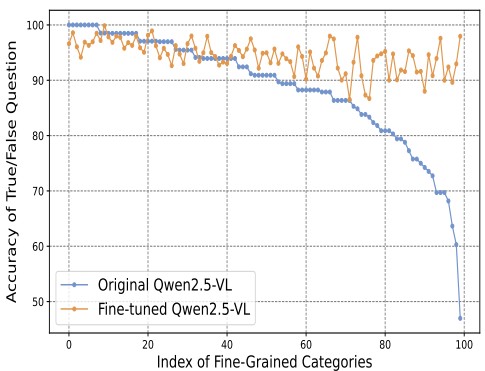

Figure 13: Comparison of the original and fine-tuned Qwen2.5-VL (Bai et al., 2025) models on occurrence-balanced fine-grained aircraft categories. True/false question accuracy for each category is ranked, with blue dots representing the original model and yellow dots the fine-tuned model.

Table 10: Attribute recognition accuracy of LLaVA (Liu et al., 2024a) on the *CUB-200-2011* (Wah et al., 2011) dataset (values in parentheses represent the average accuracy for each attribute).

| Color Attribute (44.34) | | | | | | | |
|---|---|---|---|---|---|---|---|
| belly color | 54.79 | back color | 41.90 | bill color | 41.44 | breast color | 49.56 |
| crown color | 48.71 | eye color | 69.27 | forehead color | 47.03 | leg color | 35.37 |
| nape color | 40.51 | throat color | 35.40 | under tail color | 38.88 | underparts color | 54.81 |
| upper tail color | 41.41 | upperparts color | 34.00 | wing color | 34.60 | primary color | 41.77 |
| Pattern Attribute (23.69) | | | | | | | |
| back pattern | 27.27 | belly pattern | 26.41 | breast pattern | 24.24 | head pattern | 11.35 |
| tail pattern | 23.19 | wing pattern | 29.67 | | | | |
| Shape Attribute (14.05) | | | | | | | |
| bill shape | 1.39 | shape | 18.59 | tail shape | 9.89 | wing shape | 26.34 |
| Length Attribute (15.71) | | | | Size Attribute (49.47) | | | |
| bill length | | 15.71 | | size | | 49.47 | |

## C.3 RESULTS OF ATTRIBUTE RECOGNITION

Table 2 and Table 14 shows the attribute recognition accuracy of InternVL3 and Qwen2.5-VL on the *CUB-200-2011* dataset. The results of LLaVA, BLIP2, InternVL and Gemini-2.0-flash are shown in Table 10, Table 11, Table 12, and Table 13.

Table 11: Attribute recognition accuracy of BLIP2 (Li et al., 2023) on the *CUB-200-2011* (Wah et al., 2011) dataset (values in parentheses represent the average accuracy for each attribute).

| Color Attribute (37.94) | | | | | | | |
|---|---|---|---|---|---|---|---|
| belly color | 51.15 | back color | 39.64 | bill color | 23.42 | breast color | 50.17 |
| crown color | 42.59 | eye color | 23.59 | forehead color | 43.18 | leg color | 18.77 |
| nape color | 41.55 | throat color | 53.81 | under tail color | 37.98 | underparts color | 41.60 |
| upper tail color | 37.52 | upperparts color | 33.01 | wing color | 31.25 | primary color | 33.48 |
| Pattern Attribute (11.34) | | | | | | | |
| back pattern | 14.66 | belly pattern | 7.82 | breast pattern | 9.48 | head pattern | 2.14 |
| tail pattern | 14.21 | wing pattern | 19.73 | | | | |
| Shape Attribute (25.05) | | | | | | | |
| bill shape | 8.84 | shape | 34.69 | tail shape | 13.51 | wing shape | 43.19 |
| Length Attribute (30.11) | | | | Size Attribute (27.62) | | | |
| bill length | | 30.11 | | size | | 27.62 | |

Table 12: Attribute recognition accuracy of InternVL (Chen et al., 2024) on the *CUB-200-2011* (Wah et al., 2011) dataset (values in parentheses represent the average accuracy for each attribute).

| Color Attribute (35.78) | | | | | | | |
|---|---|---|---|---|---|---|---|
| belly color | 52.09 | back color | 33.89 | bill color | 26.59 | breast color | 46.58 |
| crown color | 39.91 | eye color | 23.68 | forehead color | 40.83 | leg color | 32.75 |
| nape color | 29.66 | throat color | 30.21 | under tail color | 32.31 | underparts color | 50.57 |
| upper tail color | 33.42 | upperparts color | 29.64 | wing color | 27.17 | primary color | 40.15 |
| Pattern Attribute (34.71) | | | | | | | |
| back pattern | 35.57 | belly pattern | 44.14 | breast pattern | 42.22 | head pattern | 11.81 |
| tail pattern | 35.86 | wing pattern | 37.31 | | | | |
| Shape Attribute (23.03) | | | | | | | |
| bill shape | 12.16 | shape | 38.08 | tail shape | 15.49 | wing shape | 26.43 |
| Length Attribute (29.31) | | | | Size Attribute (47.70) | | | |
| bill length | | 29.31 | | size | | 47.70 | |

Table 13: Attribute recognition accuracy of Gemini-2.0-flash (Gemini Team, 2024) on the *CUB-200-2011* (Wah et al., 2011) dataset (values in parentheses represent the average accuracy for each attribute).

| Color Attribute (47.22) | | | | | | | |
|---|---|---|---|---|---|---|---|
| belly color | 62.09 | back color | 36.51 | bill color | 52.31 | breast color | 56.01 |
| crown color | 56.44 | eye color | 59.57 | forehead color | 53.55 | leg color | 40.66 |
| nape color | 40.40 | throat color | 60.23 | under tail color | 40.60 | underparts color | 59.65 |
| upper tail color | 39.99 | upperparts color | 29.66 | wing color | 29.21 | primary color | 38.69 |
| Pattern Attribute (56.14) | | | | | | | |
| back pattern | 56.26 | belly pattern | 70.51 | breast pattern | 66.89 | head pattern | 39.56 |
| tail pattern | 52.33 | wing pattern | 51.26 | | | | |
| Shape Attribute (48.75) | | | | | | | |
| bill shape | 61.62 | shape | 68.20 | tail shape | 32.13 | wing shape | 33.04 |
| Length Attribute (71.82) | | | | Size Attribute (52.72) | | | |
| bill length | | 71.82 | | size | | 52.72 | |

# D    MACHINE-ORIENTED EVALUATIONS

## D.1    QUALITATIVE ANALYSIS OF GRANULARITY INCONSISTENCY IN LVLM ALIGNMENT DATA

In the LVLM's alignment data, we observe a phenomenon of granularity inconsistency, where fine-grained objects in images are paired with coarse-grained textual descriptions. Figure 14 shows some examples of granularity inconsistency, as well as a constructed sample of properly aligned granularity.

In practice, ensuring fully consistent fine-grained granularity across all image-text pairs is often infeasible, especially when relying on web-scale or weakly labeled data. In our retraining experiment

Table 14: Attribute recognition accuracy of Qwen2.5-VL (Bai et al., 2025) on the *CUB-200-2011* (Wah et al., 2011) dataset (values in parentheses represent the average accuracy for each attribute).

| Color Attribute (40.39) | | | | | | | |
|---|---|---|---|---|---|---|---|
| belly color | 51.11 | back color | 32.89 | bill color | 46.50 | breast color | 44.84 |
| crown color | 46.54 | eye color | 54.85 | forehead color | 44.57 | leg color | 37.79 |
| nape color | 36.49 | throat color | 40.74 | under tail color | 34.60 | underparts color | 50.20 |
| upper tail color | 34.92 | upperparts color | 27.20 | wing color | 26.03 | primary color | 36.96 |
| Pattern Attribute (45.12) | | | | | | | |
| back pattern | 42.66 | belly pattern | 64.58 | breast pattern | 59.79 | head pattern | 14.57 |
| tail pattern | 45.04 | wing pattern | 44.11 | | | | |
| Shape Attribute (29.30) | | | | | | | |
| bill shape | 15.30 | shape | 58.17 | tail shape | 5.63 | wing shape | 38.10 |
| Length Attribute (63.20) | | | | Size Attribute (52.56) | | | |
| bill length | | 63.20 | | size | | 52.56 | |

in Table 4, we made efforts to construct more consistent alignment data, but some residual mismatch may still exist.

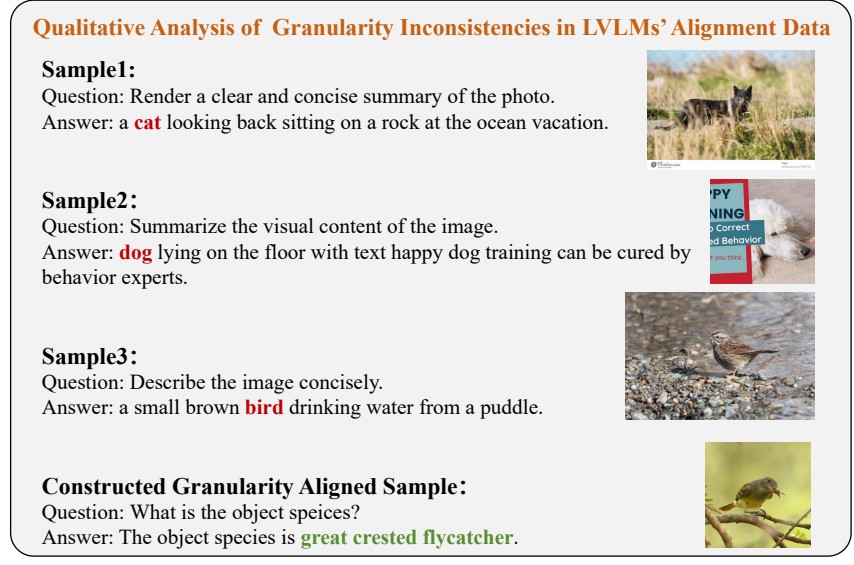

Figure 14: Qualitative analysis of granularity inconsistencies in LVLMs' alignment data and a constructed sample of properly aligned granularity.

## D.2 IMPROVING THE FINE-GRAINED DISCRIMINABILITY OF VISUAL FEATURES DURING THE ALIGNMENT STAGE CAN ENHANCE LVLM PERFORMANCE ON FINE-GRAINED TASKS.

In Table 4, we can find that the alignment strategy might impair the fine-grained discriminability of visual features. We then conduct further analysis and find that improving the fine-grained discriminability of visual features during the alignment stage can enhance LVLM performance on fine-grained tasks.

Specifically, we compare the two variants of LLaVA from Table 4 on fine-grained short-answer questions: (1) Vanilla LLaVA, where the vision-language alignment is trained on image-text pairs with granularity inconsistencies, (2) Retrained LLaVA, where the alignment module is trained on data with matched granularity.

Table 15: Linear prob classification results of LLaVA visual features and fine-tuned results of two variants of LLaVA on fine-grained short asnwer questions.

| Datasets | Linear | | | Fine-tuned | |
|---|---|---|---|---|---|
| | Origin | Aligned | Aligned-FG | Vanilla LLaVA | Retrained LLaVA |
| *CUB-200-2011* | 79.77 | 73.17 | 75.06 | 85.60 | 86.32 |
| *Stanford Dogs* | 81.24 | 78.14 | 80.69 | 86.49 | 87.58 |
| *Stanford Cars* | 87.57 | 83.90 | 85.63 | 90.55 | 91.73 |
| *Food-101* | 94.27 | 93.35 | 94.32 | 95.25 | 95.74 |

The results in Table 15 show that Retrained LLaVA consistently outperforms Vanilla LLaVA over all datasets, indicating that improving the fine-grained discriminability of visual features during the alignment stage can enhance LVLM performance on fine-grained tasks.

Building on this finding, we believe that incorporating contrastive learning objectives (e.g., patch- or region-level contrastive loss) during the alignment stage may further help preserve discriminative visual information.

### D.3    RESULTS OF CLASSIFICATION ACROSS MULTI-CATEGORIES

In Figure 8, we have shown the classification accuracy both within a single super-category and across multiple meta-categories in three datasets. Here, in Figure 15, we include more results on nine fine-grained datasets. As shown in the results, EVA-CLIP, trained with contrastive paradigm, maintains a higher score in classification across multiple meta-categories compared to Qwen and BEiT3, which are trained with generative and reconstruction paradigms.

## E    LLM USAGE DISCLOSURE

We acknowledge the use of a Large Language Model (LLM) as a general-purpose assistive tool during the preparation of this paper. Specifically, the LLM was employed in a limited capacity to refine the language, improve clarity, and polish the writing. The model did not play a significant role in research ideation, conceptualization, experimental design, or substantive content generation. All core ideas, analyses, and contributions are entirely the work of the authors.

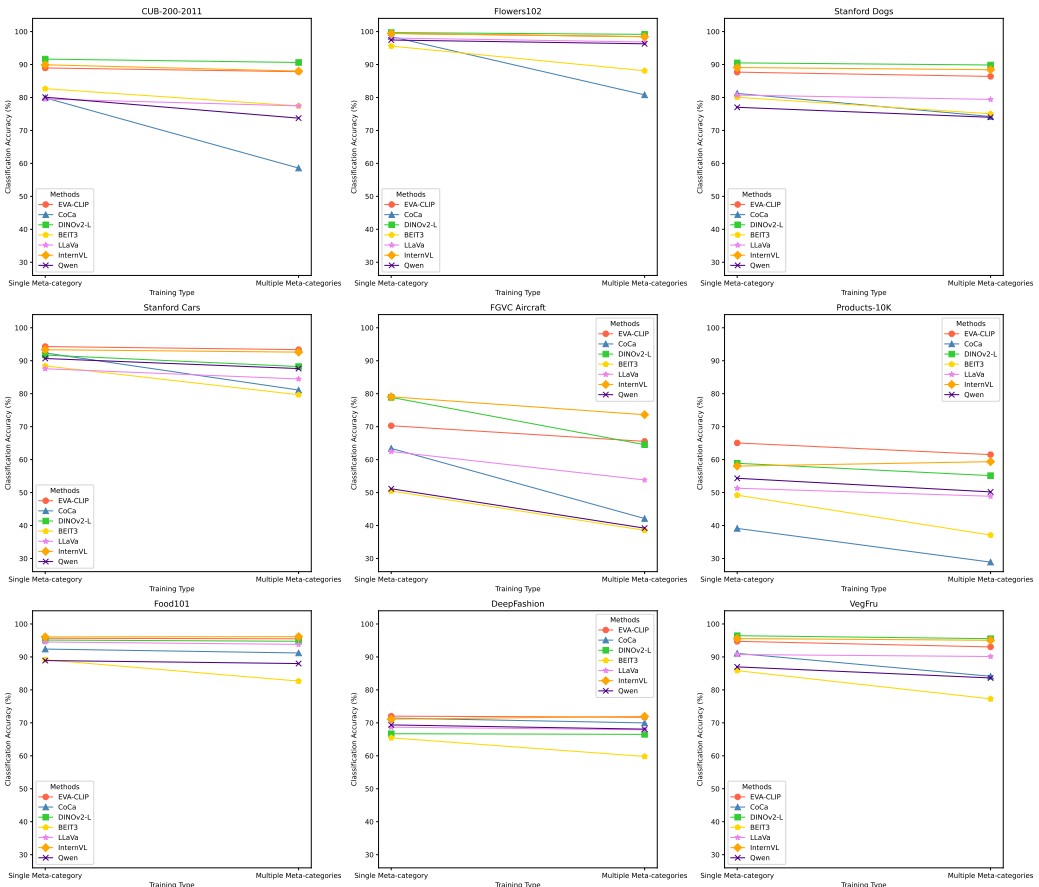

Figure 15: Classification results of LVLM visual features on fine-grained datasets. "Single" denotes accuracy from training on a single meta-category, while "Multiple" reflects accuracy from training on a unified dataset combining multiple meta-categories.

