# OpenReview forum: "Benchmarking Large Vision-Language Models on Fine-Grained Image Tasks: A Comprehensive Evaluation"
_ICLR.cc/2026/Conference — ICLR 2026 Poster_

### Official Review · Reviewer_cSv5 · 2025-10-29

**Soundness:** 3
**Presentation:** 2
**Contribution:** 3
**Rating:** 6
**Confidence:** 4

**Summary:**

This paper introduces FG-BMK, a comprehensive benchmark designed to evaluate Large Vision-Language Models (LVLMs) on fine-grained image tasks, which remain underexplored compared to generic visual reasoning or captioning tasks. FG-BMK comprises 0.28 million images and 1.01 million automatically generated questions sourced from 12 established fine-grained datasets such as CUB, Stanford Cars, and FGVC Aircraft. The benchmark features two complementary paradigms:

1. Human-oriented evaluation: measures LVLMs’ ability to answer fine-grained visual questions in conversational form (attribute recognition, hierarchical granularity reasoning, and knowledge bias estimation).
2. Machine-oriented evaluation: measures the discriminability and robustness of visual representations via retrieval and classification tasks.

The study evaluates 12 representative models (GPT-4o, Gemini-2.0-flash, Qwen2.5-VL, InternVL3, LLaVA, DINOv2, etc.) and provides systematic analyses. Main findings include:
- Contrastive training (e.g., CLIP, DINOv2) yields better fine-grained discriminability than generative or reconstruction paradigms.
- Text-image alignment can harm fine-grained distinctions, especially when text granularity mismatches image detail.
- LVLMs are more vulnerable to perturbations in fine-grained domains than generic vision tasks.
- Scaling model size or data volume brings limited gain compared to data quality and alignment design.
- LVLMs perform well on appearance description but lag behind domain-specific fine-grained models.

**Strengths:**

- Proposes a large, reproducible benchmark that systematically covers multiple aspects of fine-grained vision-language understanding.
- Provides clear empirical findings with actionable insights, notably about alignment granularity and contrastive pretraining.
- Includes both human-style and feature-based evaluations.
- Reveals interesting findings for LVLM's behaviour in fine-grained tasks.
- Gives insights for how to design and train vision representation encoders in the context of improving fine-grained perception of LVLM

**Weaknesses:**

- Primarily an evaluation paper; lacks methodological novelty or a new model proposal. I agree that a benchmarking and analysis paper can bring valuable insights to the field. However, since this paper already proposes several small fixes (fine-tuning on balanced fine-grained datasets, improved alignment strategies), could these be integrated into a more systematic approach that contributes to improving MLLMs?

- Another concern is that automatic question generation may limit linguistic diversity and realism compared to human-written prompts. There are only a few chat templates shown in Appendix A.2. Could the use of more natural prompts affect performance?

- Lack of a systematic comparison. Although this paper evaluates many mainstream models, it lacks a clear comparison of their results on FG-BMK (for example, an average of all sub-benchmarks on FG-BMK). A clear comparison would be important for readers to understand the performance and relative ranking of different LVLMs.

- The benchmark is sourced from existing public datasets, which raises concerns that different LVLMs may have been trained on overlapping portions of these datasets. This could make fair performance comparison difficult. Can the authors provide an analysis of this potential train-test overlap issue?

**Questions:**

See Weakness.

- Additionally, can the authors provide more details on the PGD attack setup for feature perturbation? L∞ PGD requires specifying an upper bound for the perturbation. Do all encoders use the same upper bound?

Overall, this paper delivers insightful analyses of LVLMs on fine-grained visual tasks, with solid and detailed evaluations from both semantic and feature-based perspectives. I recommend acceptance and would consider raising the score if the above concerns are addressed.

---

> ### Author Response · Authors · 2025-11-24
>
> **W1: Could several small fixes be integrated into a more systematic approach?**
>
> Thanks for the comment. Following your suggestion, we design a systematic approach in which, during the LLaVA pretraining stage, balanced fine-grained data are used to mitigate inconsistent recognition capabilities, and granularity-consistent data are employed to alleviate the drop in visual feature discriminability during the alignment phase. After fine-tuning on fine-grained data, the improved LLaVA achieves higher accuracy: from 79.56% to 81.32% on CUB-200-2011 short-answer questions, and from 62.28% to 64.63% on the Aircraft dataset.
>
> **W2: Concern about the template.**
>
> Thanks for the comment. To valid this, we expanded the original template to 10 diverse human-written prompts and reconstructed the multiple-choice questions in the human-oriented benchmark. As shown in Table 1 and Table 2, the template size leads to only minor changes in accuracy, and the overall LVLM behavior and observed trends remain consistent. This indicates that our tasks are robust to variations in user input.
>
> Table 1: Attribute recognition accuracy of InternVL3 using original and extended prompt on the CUB-200-2011 dataset (values in parentheses represent the average accuracy for each attribute). Accuracy are shown in the format “original / extended”, with the left representing accuracy using the original prompt and the right using the extended prompt.
>
> |Color Attribute (47.40/47.45)||||||||
> |-|-|-|-|-|-|-|-|
> |belly color|58.49 / 60.04|back color|34.98 / 36.33|bill color|51.31 / 49.64|breast color|54.25 / 55.91|
> |crown color|55.30 / 54.01|eye color|84.59 / 82.96|forehead color|53.32 / 51.90|leg color|44.01 / 45.67|
> |nape color|39.24 / 38.02|throat color|52.77 / 54.53|under tail color|34.69 / 35.80|underparts color|56.20 / 55.08|
> |upper tail color|37.30 / 38.77|upperparts color|28.75 / 27.50|wing color|30.16 / 31.88|primary color|43.05 / 41.29|
>
> |Pattern Attribute (50.13/50.28)||||||||
> |-|-|-|-|-|-|-|-|
> |back pattern|40.94 / 39.38|belly pattern|68.13 / 67.00|breast pattern|65.12 / 66.87|head pattern|35.92 / 34.66|
> |tail pattern|41.64 / 42.93|wing pattern|49.04 / 50.84|||||
>
> |Shape Attribute (30.95/31.01)||||||||
> |-|-|-|-|-|-|-|-|
> |bill shape|37.61 / 36.41|shape|52.37 / 50.60|tail shape|10.42 / 12.04|wing shape|23.39 / 24.98|
>
> |Length Attributes (71.03/69.71)||||Size Attributes (52.55/54.21)||||
> |-|-|-|-|-|-|-|-|
> |bill length|71.03 / 69.71|||size|52.55 / 54.21|||
>
> Table 2: Results of InternVL3 using original and extended prompt on true/false (TF) and multiple-choice (MC) questions across different levels of granularity on the CUB-200-2011 dataset.
>
> |TF||||||
> |-|-|-|-|-|-|
> |Class|98.79 / 98.03|Genus|85.69 / 86.19|Species|61.88 / 62.34|
>
> |MC||||||
> |-|-|-|-|-|-|
> |Class|99.42 / 99.58|Genus|88.13 / 87.62|Species|60.15 / 59.23|
>
> **W3: Lack of a systematic comparison.**
>
> Thanks for your suggestion. In the final version, we will use a radar chart, similar in style to Figure 4, to present a systematic comparison of each model's average performance across sub-benchmarks like "Attribute Recognition" and "Image Retrieval".
>
> **W4: Worry about potential train-test overlap issue.**
>
> Thanks for the comment. Given that LVLMs are pre-trained on large-scale web data and may have been trained on overlapping portions of these datasets, we consider this issue from several perspectives:
>
> First, one effective way to mitigate this is by using newer data. Recently, we have noticed that newer datasets like INQUIRE [1] actively filter out images from existing public datasets during their collection process. This step ensures that the benchmark data is less likely to be encountered during LVLM pre-training, thereby enabling a fairer evaluation.
>
> Second, when sourcing from existing datasets, a crucial practice is to only use their official test sets (as we did in our work) to minimize the risk of data leakage. Furthermore, potential data leakage can also be diagnosed by comparing a model's performance on newer datasets against established ones. For instance, if Model A underperforms Model B on newer datasets but consistently outperforms it on older benchmarks, this discrepancy might suggest that Model A's performance is inflated by data leakage on the latter.
>
> Finally, we suggest that benchmarks should be regularly maintained and updated to ensure fairer comparisons and mitigate the potential train-test overlap problem. By comparing model performance across different versions of a benchmark, we can also gain a more comprehensive understanding of how model capabilities evolve over time.
>
> [1] Edward Vendrow, Omiros Pantazis, Alexander Shepard, Gabriel Brostow, Kate E. Jones, Oisin Mac Aodha, Sara Beery, Grant Van Horn. INQUIRE: A Natural World Text-to-Image Retrieval Benchmark. Advances in Neural Inf. Process. Syst., volume37, pages 126500-126514, 2024.

---

> > ### Author Response · Authors · 2025-11-24
> >
> > **Q1: Exact setting for the projected gradient descent perturbation**
> >
> > In our experiments, we apply the standard projected gradient descent (PGD) attack by perturbing the visual features, with the perturbation direction opposite to the gradient of the loss function. Specifically, we set the perturbation step size parameter alpha to 1/255, the perturbation boundary epsilon to 0.2, and the number of iterations to 10. All encoders use the same hyperparameters.

---

> > > ### Comment · Reviewer_cSv5 · 2025-11-24
> > > **Response to authors**
> > >
> > > I thank the authors for the timely response and the additional experiments. My concerns are mostly addressed. However, I still have one remaining question regarding Q1: Are the encoder features normalized? If not, different encoders can have very different feature norms (due to affine parameters in the final LayerNorm), which makes using a uniform PGD upper bound (0.2 in your case) potentially unfair, as the *relative* perturbation strength is effectively `epsilon / ||feature||`.

---

> > > > ### Author Response · Authors · 2025-11-25
> > > >
> > > > **Are the encoder features normalized?**
> > > >
> > > > Thanks for the comment. Before answering this question, we note that our previous description unintentionally caused some misunderstanding about how the PGD attack is applied. In practice, when performing the PGD attack, we first extract the visual features of the images, then compute the loss and the corresponding gradient based on the classification results, and finally perturb the input images in the direction opposite to the gradient. In other words, the perturbation is computed from the visual features and their gradients, but it is applied to the image.
> > > >
> > > > Therefore, the encoder features are normalized by the final LayerNorm, and this does not affect the fairness of the comparison, since the perturbations are applied to the images rather than to the visual features.
> > > >
> > > > Our original words was intended to express that the perturbations are derived from the model’s visual features, and to avoid the misunderstanding that we use the same perturbed image to evaluate all models (the perturbation is related to the corresponding visual features). We will clarify the PGD setup more precisely in the revised version to avoid such confusion.

---

> > > > > ### Comment · Reviewer_cSv5 · 2025-11-26
> > > > > **Response to authors**
> > > > >
> > > > > I thank the authors for the timely and detailed explanation. My concerns are mostly addressed. I also agree with the authors that this benchmark is substantially different from existing benchmarks with similar purposes (e.g., MMVP, MERLIM), both in scale and in content. In addition, the authors provide an effective fine-tuning recipe for LLaVA that noticeably improves fine-grained accuracy.
> > > > >
> > > > > Since no other reviewers have responded yet, I will revisit my score once the broader discussion develops.

---

> > > > > > ### Author Response · Authors · 2025-11-26
> > > > > >
> > > > > > We appreciate the reviewer’s acknowledgment of our work. If there are any new questions during the broader discussion, we are glad to discuss and address them.

---

### Official Review · Reviewer_xLqa · 2025-10-29

**Soundness:** 2
**Presentation:** 3
**Contribution:** 2
**Rating:** 2
**Confidence:** 4

**Summary:**

This paper introduces FG-BMK, a large-scale benchmark designed to provide a fine-grained evaluation of Large Vision-Language Models (LVLMs) from both human-oriented and machine-oriented perspectives. The benchmark contains 1.01 million questions paired with 0.28 million images, covering 12 fine-grained visual datasets. The human-oriented evaluation focuses on assessing the model’s ability to understand and respond to fine-grained visual queries in conversational contexts. The machine-oriented evaluation measures model performance on two standard fine-grained vision tasks: image retrieval and image recognition.

**Strengths:**

1. The paper is well structured and easy to follow.

2. This paper attempts to address an important topic, the evaluation of Large Vision-Language Models (LVLMs) on fine-grained visual tasks. The authors propose a benchmark, FG-BMK, by curating questions over 12 existing fine-grained datasets.

3. It provides a large set of pair of questions-images to comprehensely evaluate LVLMs in different relevant tasks, like Attribute Recognition, Hierarchical Granularity Recognition, etc.

**Weaknesses:**

1. Limited novelty: While the topic is relevant, the paper’s novelty is limited. Several recent benchmarks, such as MERLIM (Villa et al., 2023), HallusionBench (Guan et al., 2023), MMVP (Tong et al., 2024), MMBench (Liu et al., 2024), and AMBER (Wang et al., 2023) have already evaluated LVLMs across similar tasks. The paper does not clearly explain how FG-BMK differs from or improves upon these existing benchmarks.

2. Limited model evaluation: The set of evaluated models is small and outdated. The study omits many recent (like LLaVA-OneVision, LLaVA More, EAGLE 2.5) and commercially competitive LVLMs (like, GPT-4o and Gemini).

**Questions:**

1. Novelty and differentiation: As mentioned in the first weakness, there are already numerous LVLM benchmarks covering fine-grained evaluation. Could you elaborate on the specific advantages or unique contributions of FG-BMK compared to existing benchmarks such as MERLIM (Villa et al., 2023), HallusionBench (Guan et al., 2023), MMVP (Tong et al., 2024), MMBench (Liu et al., 2024), and AMBER (Wang et al., 2023)?

---

> ### Author Response · Authors · 2025-11-24
>
> **W1 and Q1: Similar to existing benchmarks and limited novelty.**
>
> Thanks for the comment. We would like to clarify that our FG-BMK is fundamentally distinct from the benchmarks you mentioned (e.g., MERLIM, HallusionBench, MMVP). While other benchmarks mention “fine-grained” evaluation, they primarily focus on *hallucination* or *broader, general-purpose vision tasks* such as grounding and relationship understanding. In contrast, our benchmark targets *fine-grained visual tasks*, evaluating LVLMs’ ability to perceive and understand extremely similar objects—an aspect that has not been systematically addressed in prior benchmarks.
>
> To assess this ability, we carefully design tailored experiments. For example, in the hierarchical granularity recognition task, we evaluate LVLMs’ ability to leverage domain-specific knowledge to identify object categories in images across different levels of granularity within hierarchical taxonomies. In the attribute recognition task, we evaluate LVLMs’ perceptual abilities with respect to object attributes.
>
> To further illustrate the differences between FG-BMK and existing benchmarks, we outline the primary characteristics of these benchmarks below:
>
> ||Images|Questions|Evaluated Tasks|
> |-|-|-|-|
> |MERLIM|77,982|300,000|General vision tasks, recognition, grounding, relationship understanding|
> |HALLUSIONBENCH|346|1129|Visual illusion, knowledge hallucination|
> |MMVP|300|300|Scene understanding, relationship understanding, counting|
> |MMBench|3,217|3,217|Reasoning and perception, future prediction, logical reasoning, OCR, attribute comparison|
> |AMBER|1,004|15,220|Existence hallucination, relation hallucination|
> |**Ours**|**280,000**|**1,010,000**|**Fine-grained visual tasks, hierarchical granularity recognition, attribute recognition, fine-grained classification and retrieval**|
>
> **W2: Limited model evaluation and omission of recent LVLMs**
>
> Our evaluation includes 12 models and, as of the submission date, recent models such as GPT-4o, Gemini, LLaVA, InternVL3 and Qwen2.5-VL have already been evaluated in our experiments (cf. Line 199).

---

### Official Review · Reviewer_Uk4d · 2025-10-31

**Soundness:** 3
**Presentation:** 4
**Contribution:** 3
**Rating:** 6
**Confidence:** 4

**Summary:**

This paper introduces FG-BMK, a new, comprehensive benchmark for evaluating Large Vision-Language Models on fine-grained image tasks, an area the authors claim is currently underexplored. The benchmark consists of 1.01 million questions and 0.28 million images, which are sourced from 12 existing, well-established fine-grained datasets. The evaluation is structured into two paradigms, a human-oriented evaluation and a machine-oriented evaluation. Based on extensive experiments on 12 representative LVLMs, the paper reports several key findings. Critically, the authors find that contrastive training paradigms produce superior fine-grained features, and that the vision-language alignment process itself can impair fine-grained discriminability.

**Strengths:**

1. The paper's greatest strength is the FG-BMK benchmark itself. The dual-paradigm (human-oriented and machine-oriented) evaluation is effective, allowing for a holistic assessment of both conversational understanding and raw feature quality.
2. The paper delivers more than just a leaderboard. It uncovers several novel and important insights.
3. The paper is exceptionally well-written, with clear figures and tables that make the complex results easy to digest.

**Weaknesses:**

1. The "machine-oriented" evaluation provides the paper's most novel and important insights. However, this evaluation requires access to internal model features, which is impossible for closed-source models. As a result, this entire, crucial part of the benchmark cannot be applied to SOTA models like GPT-4o and Gemini.
2. The paper contains a clear factual error. The Ethics Statement explicitly claims the benchmark "does not cover sensitive areas like medical imaging," but Table 6 in the appendix clearly lists the "SkinCon" dataset, which it explicitly labels as "Dermatology".
3. The abstract's claim of "comprising... 0.28 million images"  could be mildly misleading, implying a new image collection. The core contribution is the 1.01 million questions and the evaluation framework, which are built on 12 existing public datasets.

**Questions:**

Please respond to the weaknesses I mentioned above.

**Details Of Ethics Concerns:**

The Ethics Statement explicitly claims the benchmark "does not cover sensitive areas like medical imaging," but Table 6 in the appendix clearly lists the "SkinCon" dataset, which it explicitly labels as "Dermatology".

---

> ### Author Response · Authors · 2025-11-24
>
> **W1: Closed-source models can not be evaluated on "machine-oriented" task.**
>
> Due to the inherent restrictions of closed-source models, our “machine-oriented” evaluation cannot be applied to them. Instead, we use the human-oriented evaluation, a conversational setting, to evaluate these models on fine-grained image tasks. Through carefully tailored experimental designs, we obtain several novel observations in human-oriented evaluation.
>
> Specifically, whereas prior works mostly evaluate general recognition capabilities, in our “hierarchical granularity recognition” task, we design recognition questions across different levels of granularity and find that LVLMs struggle to distinguish excessively fine-grained categories. Moreover, in the knowledge bias estimation task, our experimental design reveals that the inconsistent recognition accuracy of LVLMs across fine-grained categories can be attributed to the characteristics of their training data and the underlying LLM base, which has not been systematically analyzed in prior work.
>
> **W2: Error of “Ethics Statement”.**
>
> Thank you for your comment. We will revise the Ethics Statement in the final version to provide a detailed discussion on the ethical considerations related to the medical images used.
>
> **W3: Potential misleading of benchmark description.**
>
> Thank you for your comment. We will clarify this point in the final version to avoid any potential misunderstanding.

---

> > ### Comment · Area_Chair_p73E · 2025-11-28
> > **Request to Upload Updated Version with Ethics Statement**
> >
> > Could you please upload an updated version that includes the ethics statement? Ethical considerations need to be explicitly addressed and clearly confirmed in the submitted manuscript.
> >
> > AC

---

> > > ### Author Response · Authors · 2025-11-28
> > >
> > > Dear Area Chair,
> > >
> > > Thank you for your reminder. We have uploaded an updated version of the manuscript in which the ethical considerations are explicitly addressed and clearly confirmed in the Ethics Statement section.
> > >
> > > The Authors

---

### Official Review · Reviewer_Nmtf · 2025-11-01

**Soundness:** 3
**Presentation:** 3
**Contribution:** 3
**Rating:** 6
**Confidence:** 4

**Summary:**

This paper presents FG-BMK, a large-scale benchmark containing 3.49 million questions and 3.32 million images that systematically evaluates VLMs on fine-grained visual tasks from both human-oriented dialogue and machine-oriented representation perspectives. Across twelve domain-specific datasets, the authors demonstrate that contrastively trained encoders achieve superior subordinate-level discriminability, while generative or reconstruction-based paradigms underperform; that vision–language alignment can degrade fine-grained accuracy when image–text granularity is mismatched; and that current VLMs remain more vulnerable to adversarial perturbations and less effective at attribute-based reasoning than specialised fine-grained models, thereby identifying critical directions for future multimodal model development.

**Strengths:**

1. Introduction of FG-BMK, the first million-scale benchmark dedicated to fine-grained image understanding, offering paired human-style Q&A and machine-centric retrieval/classification protocols for comprehensive VLM assessment.

2. Extensive empirical evidence that training objectives, modality-alignment strategies, and encoder scaling choices exert measurable, task-specific effects on subordinate category recognition and robustness, with contrastive losses consistently outperforming generative or reconstruction losses.

3.Systematic disclosure of knowledge bias, granularity sensitivity, and perturbation fragility in existing VLMs, coupled with quantitative gaps relative to fine-grained expert models, providing actionable guidance for data curation and architectural refinements toward stronger fine-grained visual reasoning.

**Weaknesses:**

1. The ground-truth answers of FG-BMK are directly inherited from the original coarse-grained annotations and are paired with template-generated questions; no domain expert relabelling was performed. Consequently, a non-negligible proportion of species-level items contain incorrect negative labels or ambiguous attribute descriptions. A model may therefore be penalized for a prediction that is in fact consistent with expert knowledge, undermining the statistical reliability of the reported accuracy rankings. (e.g., in the Spotlight study of Chihuahuas, the term "Japanese spaniel" is not common knowledge, and different participants gave different answers.)

2. The machine-oriented protocol exclusively reports Top-1 accuracy and mAP, omitting recently proposed robustness and fairness measures. This conservative choice restricts the diagnostic value of the benchmark and may conceal performance disparities that become evident under more lenient or distributional metrics.

**Questions:**

It's recommended to conduct experiment below to improve the limitations in the weakness session.

1. Expert Relabelling: Commission ornithologists, botanists or other taxonomic specialists to manually verify every species-level true/false and multiple-choice pair, ensuring that labels reflect up-to-date consensus and that linguistic ambiguities are resolved.

2. Expanded Metric Suite: Complement Top-1 accuracy with Top-5 and Top-10 scores, and integrate robustness indicators such as corruption-accuracy curves or fairness-aware metrics (e.g., worst-group accuracy) to provide a richer, more nuanced assessment of model behaviour.

---

> ### Author Response · Authors · 2025-11-24
>
> **W1 and Q1: Worries about coarse-grained annotations and recommend relabelling**
>
> Thanks for the comment. Following your suggestion, we consult biology experts during the rebuttal period to relabel the species annotations in the CUB-200-2011 and Stanford Dogs datasets and re-evaluated the LVLMs on these expert-verified questions. As shown in Table 1, the re-evaluated results show only minor differences compared to our original results, indicating that our evaluation and conclusions are statistically reliable. Table 2 provides some examples of the relabeled species names. In the final version, we will apply the same relabelling procedure to the remaining benchmark data and release the updated benchmark.
>
> Table 1: Comparisons of model performance on species-level questions before and after expert relabeling. Accuracies are presented for the CUB-200-2011 and Stanford Dogs datasets in the format of “Original / Relabeled”. The “Relabeled” results are obtained using questions with expert-verified questions.
>
> |Model|CUB-200-2011 (true/false)|CUB-200-2011 (multiple-choice)|Stanford Dogs (true/false)|Stanford Dogs (multiple-choice)|
> |:-|:-:|:-:|:-:|:-:|
> |LLaVA|59.04/59.51|42.58/42.22|77.45/78.63|68.81/69.25|
> |InternVL3|62.48/62.12|61.18/61.96|92.02/91.78|93.11/92.59|
> |Qwen2.5-VL|71.49/72.09|74.04/73.32|94.50/94.89|96.74/95.97|
>
> Table 2: Examples of expert-relabeled species names.
>
> |Original species name|Relabeled species name|
> |---|---|
> |Japanese spaniel|Japanese chin|
> |Blenheim spaniel|King charles spaniel|
> |Bluetick|Bluetick coonhound|
> |Dandie dinmont|Dandie dinmont terrier|
> |Appenzeller|Appenzeller sennenhund|
> |EntleBucher|Entlebucher mountain dog|
>
> **W2 and Q2: Recommend robustness and fairness measures such as Top-5, Top-10 scores, worst-group accuracy.**
>
> Thanks for the comment. Following your suggestion, we report the Top-1, Top-5, and Top-10 scores, along with the worst-group accuracy, in Table 3.
>
> For the Top-5 and Top-10 scores, the models that lead in Top-1 accuracy remain ahead under these more lenient metrics. However, their performance margins become smaller as the metric becomes more lenient.
>
> For the worst-group accuracy, although LVLMs achieve solid recognition accuracy on average, they still fail to recognize certain fine-grained categories, likely due to the extreme long-tail distribution and high inter-class similarity inherent in datasets.
>
> Table 3: Classification results of LVLM visual features on fine-grained datasets. Metrics are shown in the format “Top-1/ Top-5/ Top-10/ Worst-Group Acc".
>
> |Model       | CUB                        | Dogs                       | Aircraft                   | Food101                    | iNat2021                   |
> |:----------|:--------------------------:|:--------------------------:|:--------------------------:|:--------------------------:|:--------------------------:|
> |EVA-CLIP    | 88.95/98.37/99.18/20.00     | 87.69/98.78/99.60/43.47     | 70.27/95.52/98.58/17.64     | 95.67/99.53/99.81/69.19     | 64.70/87.15/91.96/0.0       |
> |CoCa        | 79.89/96.78/98.53/13.33     | 81.24/97.35/98.77/50.00     | 63.40/92.50/96.70/12.12     | 92.38/99.06/99.61/63.60     | 40.59/65.63/74.76/0.0       |
> |DINOv2      | 91.65/98.72/99.32/40.00     | 90.50/98.92/99.49/50.00     | 78.88/96.57/98.61/27.27     | 95.12/99.39/99.72/70.00     | 77.07/91.69/94.39/0.0       |
> |BEIT3       | 82.67/97.48/98.94/16.67     | 80.07/97.48/99.12/38.00     | 50.47/78.63/89.79/0.0       | 89.13/98.15/99.17/46.40     | 43.55/68.19/76.15/0.0       |
> |LLaVa       | 79.54/95.42/97.73/23.33     | 80.73/97.01/98.54/48.00     | 62.46/89.79/95.25/17.64     | 94.53/99.35/99.76/72.00     | 39.77/64.58/73.09/0.0       |
> |InternVL    | 89.92/98.03/98.68/33.33     | 89.09/99.03/99.71/49.00     | 79.05/97.56/99.03/32.35     | 96.07/99.67/99.89/74.40     | 57.90/78.19/84.97/0.0       |
> |Qwen        | 80.08/96.30/98.29/16.67     | 77.02/96.11/98.19/47.27     | 51.15/81.06/89.31/3.03      | 88.90/98.63/99.24/73.60     | 38.90/64.59/72.62/0.0       |

---

### Meta-Review · Area_Chair_yrWu · 2025-12-11

**Summary:**

This paper presents a large-scale benchmark designed to evaluate fine-grained visual understanding from both human-oriented dialogue and machine-oriented representation perspectives. The authors supplement the benchmark with empirical analyses intended to offer actionable insights for improving fine-grained perception, though some findings may be viewed as expected. A majority of reviewers (3/4) acknowledge the novelty and usefulness of the proposed benchmark. Other reviewers' concerns primarily center on annotation quality, evaluation protocol design, prompt diversity, and subset evaluations, and most of these issues have been addressed in the authors’ rebuttal.

**Reviewer Concerns:**

The concerns raised span a wide range. While one reviewer questions the benchmark’s novelty, the others recognize its unique characteristics—Reviewer cSv5, in particular, remarks that the benchmark differs substantially from prior datasets. More critical concerns involve annotation quality, including calls for domain-expert relabeling, as well as potential data leakage affecting LVLM evaluation. The authors provided clarifications and additional evidence in response, which addressed most of these issues.

**Reviewer Scores:**

Reviewer Nmtf gave an initial score of 6 and is likely to maintain this score, though concerns about annotation quality remain. Reviewer Uk4d also gave a score of 6 and may keep it, with lingering concerns regarding the evaluation of machine-oriented tasks. Reviewer xLqa gave an initial score of 2, but given the benchmark novelty recognized by other reviewers, may raise the score to 4. Reviewer cSv5 indicated that most concerns have been resolved and will likely maintain the score of 6.

---

### Decision · Program_Chairs · 2026-01-26

Accept (Poster)